# Pregnancy decisions after fetal or perinatal death: systematic review of qualitative research

Eleanor Dyer [1], Ruth Bell [1], Ruth Graham [2], Judith Rankin [1]

[1]Population Health Sciences Institute, Newcastle University, Newcastle, UK
[2]School of Geography, Sociology and Politics, Newcastle University, Newcastle, UK

**Correspondence to**
Professor Judith Rankin;
judith.rankin@ncl.ac.uk

## ABSTRACT

**Objectives** To synthesise the findings of qualitative research exploring parents' experiences, views and decisions about becoming pregnant following a perinatal death or fetal loss.

**Design** Systematic review and meta-synthesis of qualitative research.

**Data sources** Medline, Web of Science, CINAHL, PsycINFO, ASSIA, Embase, PUBMED, Scopus and Google Scholar.

**Eligibility criteria** Nine electronic databases were searched using predefined search terms. Articles published in English, in peer-reviewed journals, using qualitative methods to explore the experiences and attitudes of bereaved parents following perinatal or fetal loss, were included.

**Data extraction and synthesis** Qualitative data relating to first-order and second-order constructs were extracted and synthesised across studies using a thematic analysis.

**Results** 15 studies were included. Four descriptive themes and 10 subthemes were identified. The descriptive themes were: deciding about subsequent pregnancy, diversity of reactions to the event, social network influences, and planning or timing of subsequent pregnancy. The decision to become pregnant after death is complex and varies between individuals and sometimes within couples. Decisions are often made quickly, in the immediate aftermath of a pregnancy loss, but may evolve over time. Bereaved parents may feel isolated from social networks.

**Conclusions** There is an opportunity to support parents to prepare for a pregnancy after a fetal or perinatal loss, and conversations may be welcomed at an early stage. Health professionals may play an important role providing support lacking from usual social networks.

**PROSPERO registration number** CRD42018112839

## INTRODUCTION

Perinatal and fetal death remain common adverse pregnancy outcomes,[1][2] with a perinatal mortality rate of 5.40 per 1000 births in England and Wales,[3] and estimated miscarriage rate of 25%–43%.[4][5] Many parents who have experienced fetal or perinatal loss will have further pregnancies. Debate continues about recommendations concerning the optimum interpregnancy interval following such a death.[6–9] The WHO recommends

### Strengths and limitations of this study

► This review synthesised qualitative data from primary studies describing the experiences of bereaved parents in their transition from perinatal death to pregnancy.

► We used standardised methods, including double blind screening, quality rating and data extraction.

► The themes identified were supported by evidence grounded in all included studies.

► Study participants were mainly mothers from high-income countries, with high levels of education and living with their partners, limiting the wider applicability of the findings.

► The review was limited to peer-reviewed journal articles published in English.

couples wait at least 6 months before trying to conceive again,[10] based on evidence that shorter interpregnancy intervals are associated with adverse pregnancy outcome.[9][11–13] However, a recent meta-analysis found no clear evidence to support this recommendation,[9] and UK guidance does not specify a waiting period. An additional concern is allowing sufficient time to grieve and minimise the risk of trying to replace the deceased child, both of which have been associated with psychological and bonding issues.[14–16] Hence, parents may receive contradictory advice from health professionals.[17]

Many women experience an overwhelming urge to become pregnant as soon as possible after fetal or perinatal death[18–20]; 80% of women become pregnant within 18 months of the death.[20–24] The motives and processes involved in subsequent pregnancy decisions remain unclear. Health professional involvement in the decision to conceive is encouraged, and it is important for health professionals to listen to and support women where modifiable risks can be reduced to try and avert subsequent perinatal death.[25][26] However, many women only inform health professionals once they have become pregnant,[27–29] suggesting that the consideration

of conception is primarily a personal decision between partners.[19]

This systematic review aimed to identify, appraise and synthesise existing qualitative research reporting parents' experience of the decision-making process concerning becoming pregnant again after experiencing fetal or perinatal death.

## METHODS
### Search strategy
Qualitative research reporting bereaved parents' inter-pregnancy experiences pertaining to thinking about, planning or preparing for subsequent pregnancy following a perinatal death (miscarriage, stillbirth, termination of pregnancy for fetal anomaly or neonatal death) was eligible for inclusion, if published in English in a peer-reviewed journal article. Search terms were identified using the Sample, Phenomenon of Interest, Design, Evaluation, Research Type framework[30–32] (online supplementary appendix S1).

Electronic database searches (Medline, Web of Science, CINAHL, PsycINFO, ASSIA, Embase, PUBMED, Scopus and Google Scholar) were conducted between February and June 2018. Titles and abstracts were screened by ED using the web-based tool 'Rayyan'.[33] A 10% sample was blindly double screened, and any title highlighted as potentially relevant by either reviewer was included in the full-text review. There was very high agreement between screeners (97.2%) indicating a reduced risk of screening error. Two reviewers independently read all full text articles. Uncertainties about inclusion were resolved through discussion with a third reviewer. All included studies were also manually citation searched.[34–40] Fifteen studies were included in the final review (online supplementary appendix S2).

### Quality appraisal
Study quality was assessed using the 2018 Critical Appraisal Skills Programme (CASP) checklist for qualitative research.[41] To help facilitate quality assessment, a scoring system was formulated so that numerical values were assigned to the three possible answers to the CASP questions (yes=2, can't tell=1, no=0), with a maximum possible score of 20. Studies were considered 'good quality' if the overall score was 16 or more. Two reviewers independently rated each study, and differences were resolved by discussion. The quality score was considered during data analysis to ensure that all themes were present in better quality studies. While there is a debate surrounding this approach, there is no 'gold standard' tool for critical appraisal.[42 43] As such, this pragmatic decision enabled the researchers to ascertain an indicative level of quality for all included papers which helped to mitigate the tension between reporting quality and relevance, an approach used successfully by other researchers.[44]

### Data extraction and synthesis
Data including bibliographic information, aims, method, quality assessment and findings were extracted onto an Excel form separately by two researchers to ensure consensus was reached. Qualitative data relating to first-order (participants' quotes) and second-order constructs (author interpretations, assumptions, statements and ideas) were recorded and imported into QSR NVIVO V.12 for management and thematic analysis.

Thematic analysis techniques, adapted from Butler *et al*,[38] and Harden and Thomas,[45] were used primarily by ED to systematically analyse and synthesise qualitative data from the included studies in two iterative stages:

Stage 1—coding text: data from the primary studies were examined line by line for meaning and content, and within each study conceptual codes were systematically assigned allowing emerging concepts to be aggregated across the studies. Subsequent studies were coded into pre-existing concepts, and new concepts were created when deemed necessary.

Stage 2—development of descriptive themes: the list of codes was examined and re-analysed for meaning and organised into categories. Each category was examined and descriptive themes were organised to reveal relationships and link theoretically similar sub-themes together, thus offering thought-provoking new insights into the body of knowledge in this area of research.[46]

A priori themes of interest included parents' decisions about pregnancy subsequent to loss, experiences surrounding preconception preparation and conception in the context of perinatal death, and the role of health professionals. All themes were independently checked for accuracy and reviewed by a second researcher to ensure a consensus was reached.

### Patient and public involvement
Patient and public views were not sought.

## RESULTS
### Included studies
Fifteen studies published between 1986 and 2017 and conducted in eight countries met the inclusion criteria (online supplementary appendix S2 and table S1). Sample sizes ranged from 4 to 122 participants, and fetal or perinatal deaths occurred across a range of gestations. Investigating subsequent pregnancy decisions was the main aim of five studies,[19 21 47–49] while 10 studies reported relevant findings from studies where this was not the main focus. Seven studies focused solely on women's experiences,[21 47 48 50–53] seven included both parents[15 16 19 49 54–56] and one included wider family members. Participants predominantly reported ethnicity as white,[15 16 21 49 52 56] relationship status as married or cohabiting[15 16 19 21 48–51 53 54] and had completed some form of post-secondary education.[16 48 49 54] Twelve studies were considered 'good quality', including details of ethical approval, participant recruitment, research design and evidence of researcher reflexivity (online supplementary table S2).

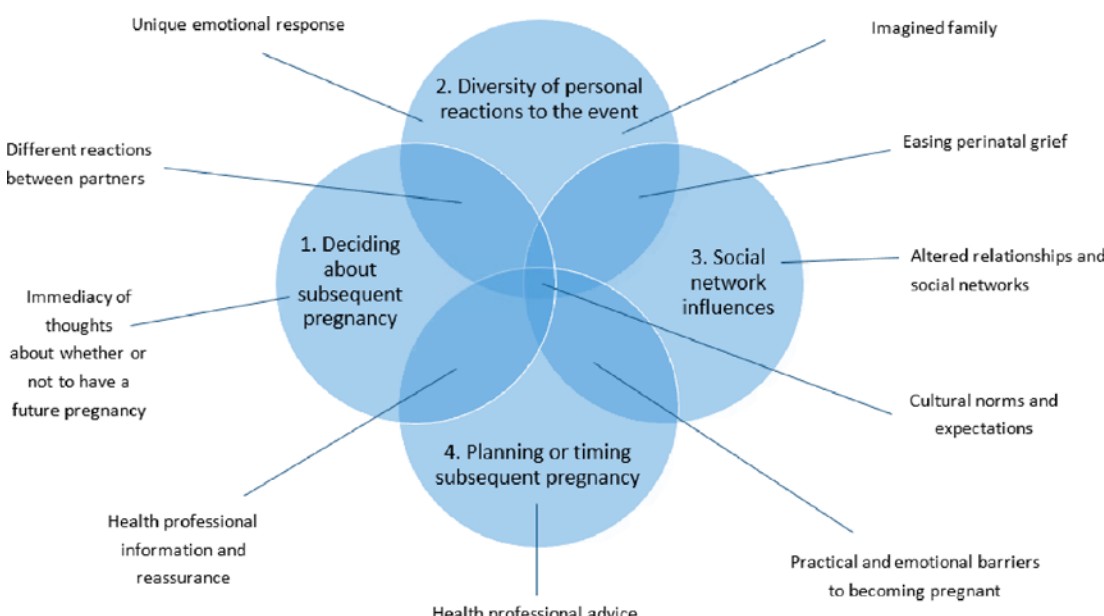

**Figure 1** Themes and subthemes

## Findings

There were four main descriptive themes: (1) deciding about subsequent pregnancy, (2) diversity of personal reactions; (3) social network influences; (4) planning or timing of subsequent pregnancy. Ten subthemes were identified which interacted with the four themes (figure 1). Themes and subthemes were tabulated according to the studies from which they arose (online supplementary table S3).

### Theme 1: deciding about subsequent pregnancy

This theme captured findings relating to the immediate period after the loss, reflecting initial thoughts about subsequent pregnancies and the point at which pregnancy planning emerges as an idea. The subthemes highlight some of the complexity involved in decision making in the aftermath of perinatal death, especially when there is disagreement between partners.

#### Immediacy of thoughts

Eight studies reported data suggesting that parents form a clear idea about whether or not they wish to try to conceive again soon after the death of their baby.[19 21 47 49–51 53 55]

> Absolutely immediately.... Whilst we were waiting to be induced, we were talking about next time.[21]

#### Differences in reactions between partners

Six studies considered the experiences of both parents; four found that parents sometimes disagreed on the decision to try for another baby.[16 19 49 54] This was a key observation in one study where fathers were observed to be more reluctant to consider a future pregnancy.[19]

> My husband was saying he didn't want any more children. So I don't know if you count this much but someone who has lost a baby, for me I would have

been pregnant coming out of the hospital again I wanted to be pregnant again that badly and then he was saying he didn't want any more at all and he wouldn't discuss it until after we got the results from the hospital.[19]

In three studies,[19 53 54] men were reported to carry a different burden to women, in particular taking on the role of 'protector' who needed to be emotionally strong,[19 53 54] and less likely to express their own fears.[19 53 54]

While there is insufficient data here to support the idea that all parents know straight away about subsequent pregnancy, this theme is noteworthy because it highlights that parents may not simultaneously feel ready, and so access to specialised care to decide what is best for them may be required soon after the event, or may be required later on. This is in keeping with the findings from Conway and Russell,[54] who emphasise the need to carry out more research into the most appropriate type and timing of intervention after loss.

### Theme 2: diversity of personal reactions

The subthemes represent how the unique and individual experiences and reactions of parents to loss is intrinsically linked in decisions surrounding the timing of subsequent pregnancy which in turn influenced how each individual engaged with the concept of future pregnancy decisions.

#### Emotional response

Emotional responses to the death of a baby are unique to each parent, and varied from feelings of self-blame and guilt, loneliness and emptiness, anger, fear, failure and shame, to sadness and grief. In turn, the emotional response may not only influence, but also shape, understandings of the self and identity. In turn, these impact decisions about becoming pregnant after loss. In four

studies, participants described the death as a void that could only be filled by a subsequent pregnancy.[15 21 47 55]

> Emotionally everything had been geared towards having a baby and then there was a big hole, a baby-shaped hole which was much bigger than a baby.[21]

### *Imagined family*

Findings from 11 studies highlighted how pregnancy and childbirth are conceptualised within individuals' life narratives, and how parents often have pre-existing expectations about their reproductive aspirations. Experiencing a perinatal death prompts parents to reconsider their life goals.[16 19 51]

> …but I was 29 when she was born, and I had this vision in my head that I was going to have all my kids by the time I was 30, and that wasn't working out, and so we better get on….[16]

### *Easing perinatal grief*

Five studies[21 47 49 51 55] reported that, for some parents, a subsequent pregnancy was considered essential for their recovery, eased feelings of perinatal grief and gave hope for the future.

> I knew it was going to be a long process and it helped me recover because it made me focus on the future and I was more hopeful.[49]

However, not all parents were so certain that a subsequent pregnancy would aid recovery, and needed longer to overcome the grief and feel ready.[19 21 47 49] These framings set the scene for future pregnancy planning which emerged and developed with reference to how an individual has interpreted the meaning of the prior loss in relation to their sense of self.

This theme highlights just how individual and personal the decision to become pregnant is, and helps to make sense of why parents can react so differently to similar circumstances of loss. This necessarily colours the context in which questions about future pregnancy planning emerge, and develop, in ways that are highly dependent on how an individual has interpreted the meaning of the previous incidence of loss in relation to their sense of self.

### Theme 3: social network influences

Future pregnancy planning decisions could be influenced by an individual's social network and the cultural norms and expectations within their social group. Comments and assumptions by others about future planning could be interpreted as supportive or distressing, depending on the context.

### *Altered relationships and social networks*

Eight studies[16 19 48 50 52 55–57] highlighted how relationships and social networks changed after a perinatal death. In three studies, some parents reported much support thereby reinforcing bonds with their social network.[21 48 56] However, in five studies, parents felt that

friends and family were not supportive, and relationships were profoundly altered following unhelpful attitudes and responses.[19 48 50 52 56] As a result, bereaved parents may withdraw or isolate themselves from friends, family and colleagues to avoid hurt,[19 48 50 52 56] and avoid discussing their loss to prevent their friends and family feeling uncomfortable.[56]

> Everything is okay with my friends, as long as I don't talk about my baby. When I do, they look away and change the subject. What am I supposed to do? I need to talk![56]

> [I] talked a lot about it with my boyfriend and friends. On the one hand it was fine, on the other hand not. None of my friends have experienced this, so it is quite difficult for them to understand. And then sometimes they said…well at least you know that you can become pregnant…I got that kind of remark.[48]

This subtheme highlights that the concept of subsequent pregnancy planning may be complicated by the comments and assumptions made by other within an individual's social network some of which could be interpreted as supportive, but in some circumstances, distressing.

### *Cultural norms and expectations*

Nine studies discussed the impact of cultural norms on decisions about subsequent pregnancy.[16 19 48 50–53 55 56] Parents in four studies noted other people's discomfort, reporting that such uneasiness led to unsupportive reactions.[16 21 52 56]

In some cultures, while there are strong expectations for women to bear children, stillbirth is regarded as taboo and parents are not expected to see or to discuss their dead baby.[50 51] A striking finding was the guilt and sense of failure that was reported following a perinatal death in this environment.[50 51] This led some women to pursue pregnancy as soon as possible to undo the sense of wrongdoing associated with stillbirth[50] and fulfil the cultural expectations of being a successful mother.[51]

> I am trying to have a baby again. As long as I can have a baby, my mother-in-law can't look down on me. Also, failing to deliver a baby makes me feel that I am not a woman. Every woman should be able to deliver a baby—that's what makes a woman so different from a man.[50]

This theme highlights that the concept of subsequent pregnancy planning may be complicated by the comments and assumptions made by others within an individual's social network. Pregnancy and childbirth may be viewed by society as a natural and celebratory part of life, rather than the anxiety-filled prospect that bereaved parents now face. Some comments could be interpreted as supportive, but some parents may perceive the expectation for them to 'get over it' and 'get on with it' hard to tolerate, especially as some felt that a subsequent pregnancy could be dismissive of their deceased baby.

### Theme 4: planning or timing subsequent pregnancy

Encapsulated in this theme is the notion that planning or timing of a subsequent pregnancy may be influenced by the advice, information and reassurance from health professionals which may differ depending on the emotional and medical barriers faced by bereaved parents after loss.

Eleven studies reported discussion of planning or timing a subsequent pregnancy.[16 19 21 47–49 52 54–57] In six studies, parents felt that the timing of a subsequent pregnancy should be based on personal reasons and individual experience, rather than the thoughts of others, including medical professionals.[19 21 47 49 52 56] Women spoke of listening to their body and trusting their own feelings.[21 47] Planning a subsequent pregnancy often involved overcoming barriers,[16 47 49 55 57] and sometimes conflicting messages from health professionals.[56]

#### Barriers: practical and emotional

In five studies, parents cited a range of emotional and medical challenges as barriers to conceiving after loss.[16 47 49 55 57] Parents faced a period of ambiguity and uncertainty when planning their next pregnancy,[16 19 21 48 49 55 57] and a loss of control associated with a long conception period.[48]

> The last time I was actually very impatient because after that last miscarriage it lasted a year and a half before we were pregnant again. So I thought it would take a year and a half again to become pregnant so we tried again a month after the miscarriage.[48]

Such an uncertainty was intensified by practical factors, such as the number of previous perinatal deaths experienced, advanced maternal age, financial strain, parent relationship status and fertility problems.[48 49 57] High levels of anxiety and fear were linked to the lack of reassurance that a subsequent pregnancy would end successfully.[16 19 52 54 55 57] Such barriers may redefine decisions about how many children parents wish to have,[57] or how soon parents try to conceive, with some deciding to delay to make sure they felt emotionally and physically ready for pregnancy,[21 47–49 55] whereas others may try to become pregnant regardless of emotional or physical readiness for fear of not being able to conceive again.[16]

> Mother: I also had a real fear of not becoming pregnant again, I thought that maybe that was my once in a lifetime shot. So that was on my mind too, that maybe I wouldn't get pregnant. So after 6 months I was real anxious to get going because I thought it might take a while.
>
> Father: Well, we missed one month, and of course right away L. panicked and figured out she'd never become pregnant again.[16]

#### Health professional advice

Seven studies discussed the involvement of health professionals in the interpregnancy period. The amount and type of advice given varied, but included answering medical questions, providing information and advice about subsequent pregnancy and reassurance about the level of care that would be received for a subsequent pregnancy.[16 19 47–49 54 56]

One early study focused specifically on mothers' perceptions of medical advice about timing of a subsequent pregnancy.[47] Again, the advice varied; five women were advised to wait less than 6 months before trying to conceive, matching the women's expectations, as they felt an urgent need to get pregnant.[47] Fourteen were advised to wait at least 6 months, 11 of whom found this advice unacceptable mainly due to the strong desire to have a baby:

> [12 months] is an eternity…I had all this parenting energy and nowhere to direct it….[47]

Five women in this study received no *specific* advice about waiting. These women appreciated this both at the time and in hindsight, as it empowered them to make their own informed decision based on their individual needs. This was a consistent finding in other studies[21 47 49 56] and highlighted the importance of *appropriate* advice, rather than the amount or the content of the advice.

#### Health professional information and reassurance

Concerned parents sought reassurance and information from health professionals, for example, about the risks of another negative outcome and the type of specialised care and emotional support they might expect to receive in a future pregnancy.[16 19 21 47 49 52 55 56]

> …they basically told us that we had the same chance of it happening to us again as if it had never happened to us. The same as any couple walking down the street. But the only thing I kept saying was that it did happen to us you know so they can't give any reassurances.[19]

However, not all parents received such advice and support,[21] and although parents can be informed of level of risk, it was not possible to guarantee that there will not be another negative outcome in a future pregnancy.[19]

This theme highlights how bereaved parents may turn to health professionals for advice, information and reassurance regarding the timing of a subsequent pregnancy and overcoming barriers to conception.

## DISCUSSION

This review analysed and synthesised qualitative data from primary studies, describing the experiences of bereaved parents in their transition from perinatal death to pregnancy, an area largely overlooked in the literature to date. The sample included a range of experiences from both parents, across eight countries, and included parents who had experienced different types of fetal or perinatal death. The themes identified were supported by evidence grounded in the primary studies. This systematic review

followed the Enhancing Transparency in Reporting the Synthesis of Qualitative Research reporting structure (online supplementary appendix S3). To reduce the risk of bias, we used standardised and comprehensive methods, with explicit criteria, double blind screening, quality rating and data extraction. As researchers, not practitioners in the field, we have a particular interest in understanding both lay and professional experiences of healthcare provided in the context of distressing life events such as reproductive loss. It was therefore important to stay close to the data so as not to impart our own opinions or judgements on the findings.

Many parents decide whether or not they wanted to have another pregnancy shortly after the loss. However, parents sometimes disagreed with their partners and early decisions could evolve over time. A key theme was the range of unique and personal reactions to loss. These experiences were intrinsically linked to decisions surrounding the timing of subsequent pregnancy which in turn influenced how participants engaged with future pregnancy decision-making. Future pregnancy planning may be influenced by an individual's social network and the cultural norms that guide people within their social group. The taboo that surrounds perinatal death makes it a difficult subject to discuss, potentially leaving bereaved parents feeling isolated and lonely. A salient finding was the assertion that parents should be provided with information about timing subsequent pregnancy, rather than prescriptive advice or specific recommendations.[21 47 49 56] Health professionals need to be mindful of the patients' individual preference for the amount and type of advice that they want or need. Parents should be empowered to access information at a time of their choosing, when they feel ready.

Resource constraints limited the review to journal articles published in English. Relevant material was identified in other sources including abstracts and theses, and some themes may not have been identified owing to the large number of search results and the fact that relevant studies may report data as secondary findings, increasing the likelihood of screening error resulting in studies being missed. The majority of study participants were mothers; fathers' experiences may be underrepresented. All studies were undertaken in high-income countries, and participants were mainly educated and living with the father of the baby. This has implications for the wider applicability of the findings.

There are inherent limitations associated with systematically reviewing qualitative studies, since only data which has been selected for presentation within the study is available, rather than the totality of the data collected. The quality of a systematic review further depends on the quality of the studies it includes. While the majority of the included studies were rated as good quality, not all studies indicated the data analysis technique used or the retrospectivity of loss which may impact the reliability of findings. Care was taken to ensure that themes were represented across multiple studies, one study rated as average quality contributed to six of the ten subthemes and dominated the findings in the subtheme regarding health professional advice. Thus, caution should be exercised regarding the wider applicability of this theme.

The findings suggested that some parents develop a clear idea soon after perinatal death whether or not they wish to pursue a subsequent pregnancy. This reaction to pregnancy loss may reflect a strong desire to leave the liminal phase that parents experience following the death of a baby, whereby the nebulous identity of becoming non-pregnant leaves parents stranded between the stable states of being pregnant and parenthood.[56 58] Other explanations include the theory that mothers and fathers assume their role of parents early in pregnancy,[59] with psychological preparation for parenthood beginning before conception.[60] Jaffe and Diamond further suggest that people determine their own 'reproductive stories' as early as their own childhood; the perinatal death leaves this desire unfulfilled until a living child is born.[61] While there is insufficient data to conclude that all parents decided quickly about subsequent pregnancy, we suggest that some parents may welcome opportunities to discuss their plans earlier than health professionals may assume. Conway and Russell emphasise the need to understand the most appropriate type and timing of health professional intervention after loss.[54]

The data describe a range of reactions to loss, and highlight the personal and individual nature of timing a subsequent pregnancy. This is consistent with other studies.[18 62–64 64–68] Our findings suggest that parents may feel increased isolation after fetal or perinatal death, due to the taboo nature of the topic.[18 69 70] Social withdrawal has also been identified as an expression of grief following bereavement.[71 72]

The needs of bereaved parents during the period when they are contemplating becoming pregnant again has received substantially less interest than the need for additional support during subsequent pregnancy. Several studies have explored the experiences of parents during pregnancy following loss, and services have been developed to respond to their need for increased support. The process of conceiving the subsequent pregnancy is not discussed in these reports.[22 28 73–75] It is possible that health professionals do not consider involvement in such decisions as part of their role. However, there are a number of causes or risk factors for fetal and perinatal death which may be amenable to intervention in the interpregnancy interval, for example, diabetes, obesity or smoking.

This review highlights that the window of opportunity to influence these factors and support parents in reducing the risks in the next pregnancy may be relatively short. Health professionals may play an important role as a bridge between medical information and emotional support in the interpregnancy interval, but with a paucity of research looking specifically at the perspectives of health professionals it is not possible to establish why advice may differ between health professionals and

whether they feel able to offer the individualised advice and support that bereaved parents require or seek.

## CONCLUSION

Many parents think about becoming pregnant again very soon after experiencing a pregnancy loss, and health professionals should anticipate the need to facilitate conversations from the very earliest point. Providing personalised and flexible support may be challenging for health professionals working within healthcare systems which may not easily adapt to differing needs. Health professional perspectives are currently analytically underdeveloped in the literature, so should be an area for further development. Further research should also address the need to better understand the decision-making process, support requirements and challenges that parents face in the interpregnancy interval following pregnancy loss, and address how bereavement services can integrate better with pregnancy preparation services. This could be of particular benefit to those with medical conditions, such as pre-existing diabetes, where effective preparation for pregnancy and delayed conception have been shown to significantly improve outcomes.[76 77]

**Contributors** ED contributed to protocol development, performed the searches, reviewed the papers, conducted the thematic synthesis and wrote the first draft of the article. RB developed the idea for the study, contributed to protocol development, reviewed papers for inclusion, quality assessment and data extraction, and contributed to writing the final article. RG developed the idea for the study, contributed to protocol development, reviewed papers for inclusion, quality assessment and data extraction, supervised and reviewed the thematic synthesis, and contributed to writing the final article. JR developed the idea for the study, contributed to protocol development, reviewed papers for inclusion, quality assessment and data extraction, and contributed to writing the final article.

**Funding** This work was supported by the Economic and Social Research Council [grant number ES/P000762/1].

**Competing interests** None declared.

**Patient consent for publication** Not required.

**Provenance and peer review** Not commissioned; externally peer reviewed.

**Data availability statement** All data relevant to the study are included in the article or uploaded as supplementary information.

**ORCID iDs**
Eleanor Dyer https://orcid.org/0000-0002-4797-1459
Ruth Bell http://orcid.org/0000-0002-8464-8933
Ruth Graham https://orcid.org/0000-0001-9432-760X
Judith Rankin https://orcid.org/0000-0001-5355-454X

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
