## [Reviewer comments · BMJ Open]

ARTICLE DETAILS

TITLE (PROVISIONAL)	Pregnancy decisions after fetal or perinatal death: systematic review of qualitative research
AUTHORS	Dyer, Eleanor; Bell, Ruth; Graham, Ruth; Rankin, Judith

VERSION 1 - REVIEW

REVIEWER	Mike Rennoldson Nottingham Trent University, UK
REVIEW RETURNED	15-Mar-2019

GENERAL COMMENTS	I found this to be a well designed, thoroughly reported, review on a focussed topic of clinical importance. It is hugely challenging to report a review of this volume of qualitative research within this word count, and yet it is important to do so to ensure findings reach a wide audience and I congratulate the authors on their efforts and achievements here. I am more knowledgeable about systematic review methods rather than the topic, so focus my comments on methods. The review was pre-registered part-way through, and the report follows the procedures outlined in the registration. The search strategy is well reported, and prioritised sensitivity - a very large number of search results were screened. The steps for quality appraisal are well explained, as is the process for data extraction and synthesis. The analysis itself strikes an effective balance between depth and breadth, is well illustrated by primary data extracts, and is plausible and useful. It is helpful to have the assurance of the representation of themes across included studies. The discussion usefully discusses the findings in the light of relevant theory and practice (although I am less familiar with this area, so not so well placed to judge this), and is transparent about strengths and weaknesses of the review. Suggested issues for the authors to address: 1. Given such a large number of search results, and the fact that relevant studies may report data as secondary findings, there is a high likelihood of screening error resulting in studies being missed. I would advise reporting this as a potential weakness of the study - I suspect the screening of 10% of results by a second author isn't a sufficient guard against error here unless there was very high agreement between screeners.2. I am not sure that using score cut-offs on the CASP tool is justified. The rubric on the tool specifically advises against this, and no reference is given to support the use of a cut-off and the
--

	particular level. I understand there is a debate on this question, but there are important doubts about such a reduction of judgements of quality, especially in qualitative research. Can the authors justify this decision, or perhaps offer a descriptive overview of the strengths and weaknesses of the literature? 3a. The table summarising studies doesn't mention the data analysis method, I think this is an important issue to help judge the kinds of data that are being synthesized. b. In the table I think the time post- fetal or perinatal death that data collection took place is a critical issue and should be highlighted for each study. c. The reporting of study characteristics data could be improved in the table. I think it is driven by the varied reporting in the studies, perhaps quoting the studies verbatim... but it is quite confusing for the reader. For example in 'Conway and Russell' what does a 48% open ended questionnaire mean? Phipps is described as a 'self-reported interview'. I wonder whether it might be more helpful for the reader for some more standardised characterisation of the studies here? The key characteristics of participants column could also do with similar attention. 4. In the analysis the report theme 2 could do with further work. a. Emotional response is described as unique to each parent, but very little variability is reported - can a little more detail of variability be given? b. The sub-theme descriptor 'aspirations for a family' doesn't quite convey the content of the very interesting sub-theme (a revision of projections of one's future). 5. The theme on health professional advice was the weakest in the results - the real value in qualitative research for me is in identifying connections and processes, but this theme is a little too descriptive (eg. listing the advice given to participants). 6. In the discussion there is some material that should really appear earlier (eg. reporting the aims of studies). 7. Do you have any suggestions for future research?
--	--

REVIEWER	Margaret Bearman Deakin University Australia
REVIEW RETURNED	27-Mar-2019

GENERAL COMMENTS	My expertise is in qualitative synthesis therefore my comments mostly pertain to methodology rather than the specific area of the systematic review. Overall the review appears cohesive and well considered, with appropriate methods choices within the systematic review tradition. I have several suggestions for improvement, mostly revolving around the point of synthesis. Firstly, I think that the research aim sells the study short. It currently is to "identify,, appraise and summarise". I think that qualitative synthesis does more than summarise, it synthesises. The authors do this to some degree but this is where I think there is room to move and an explicit statement of this in the aim would be welcome.
---

	One way to promote synthesis is to look at theme titles. Some of the themes titles (1, 2 and 4 particularly), reflect clusters of ideas rather than an overall theme. The theme "personal reactions" really didn't seem to encompass the subthemes which were about grief, loss and changing aspirations. I would have liked to see further comment about what the synthesis of these notions were and how the theme itself could have encapsulated a stronger qualitative sense of its parts. I think this would then enable the discussion to be sharper. I feel less able to comment on the discussion as I am a methodologist rather than a researcher in this particular arena. However, in my experience, by focussing on what unique features the review reveals through the sum of its parts is really what makes its contributions. (If the authors are interested, I make this point in: Quality and literature reviews: beyond reporting standards https://doi.org/10.1111/medu.12984). On a methodological note, I wanted to know how the primary and secondary constructs were combined. It appears that only the primary constructs were coded? How many researchers coded? Was there any consensus? More detail would be helpful. I would also welcome some reflexive considerations - knowing the stance and interest of the researchers would also be useful.
--	---

REVIEWER	Antje Horsch University of Lausanne and Lausanne University Hospital, Switzerland
REVIEW RETURNED	21-Jul-2019

GENERAL COMMENTS	Thank you for inviting me to review this important and timely systematic review, which will be helpful to researchers and clinicians alike. This is a rigorous and well-written manuscript but I have a few comments that I would ask the authors to address before publication. Results Theme « altered relationships and networks »: the link to decisions about subsequent pregnancies does not seem to be clearly grounded in the data but more an assumption of the researchers ? Could you please provide another quote showing this link? Theme "Barriers emotional and physical": can you please expand more on the sense of emotional readiness for a subsequent pregnancy? "Five women in this study received no specific advice about waiting. These women appreciated this both at the time and in hindsight, as it empowered them to make their own informed decision based on their individual needs. This was a consistent finding in other studies^{21, 43, 51, 52} " This is in contrast to the aim of your study enabling healthcare professionals to give advice to women related to the subsequent pregnancy after perinatal loss. Can you please discuss this?
--

	Discussion “A salient finding was the assertion that parents should be provided with information about timing subsequent pregnancy, rather than prescriptive advice or specific recommendations^{21, 43, 51, 52.}” This sentence seems in direct contrast to the following sentence in your results: “Five women in this study received no specific advice about waiting. These women appreciated this both at the time and in hindsight, as it empowered them to make their own informed decision based on their individual needs. This was a consistent finding in other studies^{21, 43, 51, 52}”. “The themes identified were supported by evidence across all 15 studies.” This does not seem to be reflected in your Table S3, e.g., the theme “social network responses” is not supported by all studies.
--	---

VERSION 1 – AUTHOR RESPONSE

Reviewer: 1

Mike Rennoldson, Nottingham Trent University, UK

Reviewer Comments	Author Response	Page/Line Number in Marked Copy
1. Given such a large number of search results, and the fact that relevant studies may report data as secondary findings, there is a high likelihood of screening error resulting in studies being missed. I would advise reporting this as a potential weakness of the study. I suspect the screening of 10% of results by a second author isn't a sufficient guard against error here unless there was very high agreement between screeners.	The level of agreement between the authors was very high at 97.2%. This has been added to search strategy. A sentence has also been added to the discussion to refer to this as a potential weakness.	Page 4, line 12-13 Page 14, lines 4-6
2. I am not sure that using score cut-offs on the CASP tool is justified. The rubric on the tool specifically advises against this, and no reference is given to support the use of a cut-off and the particular level. I understand there is a debate on this question, but there are important doubts about such a reduction of judgements of quality, especially in qualitative research. Can the authors justify this decision or perhaps offer a descriptive overview of the strengths and weaknesses of the literature?	We acknowledge that there is a current debate surrounding this approach and have added a justification and references to support our decision to use cut-offs.	Page 4, lines 26-30
3a. The table summarising studies doesn't mention the data analysis method I think this is an important issue to help judge the kinds of data that are being synthesized.	3a. The data analysis method has now been added to Table S1. Not all studies specified their data analysis method and this has been added as a point in the discussion.	Table S1 Page 14, lines 15-16

Reviewer Comments	Author Response	Page/Line Number in Marked Copy
b. In the table I think the time post- fetal or perinatal death that data collection took place is a critical issue and should be highlighted for each study.	3b. Table S1 has been updated to provide the timing of the loss.	Table S1
c. The reporting of study characteristics data could be improved in the table. I think it is driven by the varied reporting in the studies, perhaps quoting the studies verbatim... but it is quite confusing for the reader. For example in 'Conway and Russell' what does a 48% open ended questionnaire mean? Phipps is described as a 'self-reported interview'. I wonder whether it might be more helpful for the reader for some more standardised characterisation of the studies here? The key characteristics of participants column could also do with similar attention.	3c. The terminology in the data collection method column in Table S1 has been updated. The key characteristics of participants column has been updated, however, we made the decision to include key characteristics verbatim as there was so much variation between the studies it was not possible to standardise and we did not want to lose important detail.	Table S1 Table S1
4. In the analysis the report theme 2 could do with further work. a. Emotional response is described as unique to each parent, but very little variability is reported - can a little more detail of variability be given? b. The sub-theme descriptor 'aspirations for a family' doesn't quite convey the content of the very interesting sub-theme (a revision of projections of one's future).	4a. Additional detail of variability of emotional response has been added. 4b. We have changed the title of this sub-theme to 'Imagined Family' as we feel this better captures the content.	Page 7, lines 31-33 Page 8, line 7 Table S3 and Fig 1 also updated)
5. The theme on health professional advice was the weakest in the results - the real value in qualitative research for me is in identifying connections and processes, but this theme is a little too descriptive (eg. listing the advice given to participants).	We agree that this theme is a little descriptive mainly because health professionals' perspectives were under analytically developed in the primary studies. We have highlighted that this area is ripe for further development and suggested it as an area for future research.	Page 15/16, line 34/1
6. In the discussion there is some material that should really appear earlier (eg. reporting the aims of studies).	Reporting of the aims of studies has been moved from the discussion to the results section	Page 5/6, lines 33-34/1
7. Do you have any suggestions for future research?	Further future research suggestions have been added to the conclusion.	Page 15/16, line 34/1

Reviewer: 2

Margaret Bearman, Deakin University, Australia

Reviewer Comments	Author Response	Page/Line Number in Marked Copy
Firstly, I think that the research aim sells the study short. It currently is to "identify,, appraise and summarise". I think that qualitative synthesis does more than summarise, it synthesises. The authors do this to some degree but this is where I think there is room to move and an explicit statement of this in the aim would be welcome.	Thank you for highlighting this. The research aim has been amended accordingly.	Page 3, line 28
One way to promote synthesis is to look at theme titles. Some of the themes titles (1, 2 and 4 particularly), reflect clusters of ideas rather than an overall theme. The theme "personal reactions" really didn't seem to encompass the subthemes which were about grief, loss and changing aspirations. I would have liked to see further comment about what the synthesis of these notions were and how the theme itself could have encapsulated a stronger qualitative sense of its parts.	A sentence has been added to the beginning and end of each theme to better describe how/why the subthemes were encapsulated within each theme.	Pages: 6, lines 19-20 7, lines 17-22 / 25-27 8, lines 29-33 9, lines 25-28 10, lines 12-18 / 21-24 12, lines 15-16 / 31-33
I think this would then enable the discussion to be sharper. I feel less able to comment on the discussion as I am a methodologist rather than a researcher in this particular arena. However, in my experience, by focussing on what unique features the review reveals through the sum of its parts is really what makes its contributions. (If the authors are interested, I make this point in: Quality and literature reviews: beyond reporting standards https://doi.org/10.1111/medu.12984).	Thank you for sharing this helpful article, which has been added and referenced. We have amended the discussion to make it clearer what has been added from this synthesis	Page 5, lines 17-18
On a methodological note, I wanted to know how the primary and secondary constructs were combined. It appears that only the primary constructs were coded?	Both primary and secondary constructs have been coded. The findings were predominantly first-order constructs, but second-order constructs (i.e. the interpretations or conclusions of the authors) were used to support the findings. Data extraction and synthesis have been	Page 5, lines 10, 12-13

Reviewer Comments	Author Response	Page/Line Number in Marked Copy
	updated to make this clearer.	
How many researchers coded? Was there any consensus? More detail would be helpful	Additional detail has been added to make this clearer.	Page 5, line 8
I would also welcome some reflexive considerations - knowing the stance and interest of the researchers would also be useful.	We tried to stay as close to the data as possible, however, in the interest of transparency, some reflexive considerations have been added to the discussion.	Page 13, lines 10-13

Reviewer: 3

Antje Horsch, University of Lausanne and Lausanne University Hospital, Switzerland

Reviewer Comments	Author Response	Page/Line Number in Marked Copy
Theme « altered relationships and networks »: the link to decisions about subsequent pregnancies does not seem to be clearly grounded in the data but more an assumption of the researchers? Could you please provide another quote showing this link?	A quote and supporting sentence have been added.	Page 9, lines 20-23
Theme “Barriers emotional and physical”: can you please expand more on the sense of emotional readiness for a subsequent pregnancy?	We have expanded on this and added another quote.	Page 11, lines 21-23 / 25-30
“Five women in this study received no specific advice about waiting. These women appreciated this both at the time and in hindsight, as it empowered them to make their own informed decision based on their individual needs. This was a consistent finding in other studies ^{21, 43, 51, 52} “ This is in contrast to the aim of your study enabling healthcare professionals to give advice to women related to the subsequent pregnancy after perinatal loss. Can you please discuss this?	We agree that this could have been explained more explicitly. This is about parents receiving appropriate advice from healthcare professionals; not so much about the content of the advice given, but when and how much. It is a message that healthcare professionals need to listen and support, rather than give prescriptive advice. They need to be mindful of individual preference for the amount of advice sought and parents need a choice on when and how such information is accessed. It is about empowering parents to make a choice to access information at a time of their choosing rather than about the	Page 12, lines 15-16 / page 13, lines 24-26

Reviewer Comments	Author Response	Page/Line Number in Marked Copy
	healthcare professionals giving out advice or recommendations when parents are not ready for it. There is a difference between telling parents how long they should wait and giving information about the evidence in relation to child spacing. This has been amended in the subtheme and discussion to better reflect this nuance.	
Discussion “A salient finding was the assertion that parents should be provided with information about timing subsequent pregnancy, rather than prescriptive advice or specific recommendations ^{21, 43, 51, 52.} ” This sentence seems in direct contrast to the following sentence in your results: “Five women in this study received no specific advice about waiting. These women appreciated this both at the time and in hindsight, as it empowered them to make their own informed decision based on their individual needs. This was a consistent finding in other studies ^{21, 43, 51, 52} ”.	This is similar to the point above and we agree that it could have been explained more explicitly that ‘information’ is not the same as ‘advice’. Further clarification of this nuance has been added to the text.	Page 13, lines 24-26
“The themes identified were supported by evidence across all 15 studies.” This does not seem to be reflected in your Table S3, e.g., the theme “social network responses” is not supported by all studies.	This sentence has been clarified to reflect that the evidence was grounded in data across all primary studies.	Page 2, line 32

VERSION 2 – REVIEW

REVIEWER	Mike Rennoldson Nottingham Trent University, UK
REVIEW RETURNED	21-Nov-2019

GENERAL COMMENTS	Thank you for the clear and thorough response to my review. I think you have addressed all of my points well.
---